# Preparation and Characterisation of Activated Carbon from Palm Mixed Waste Treated with Trona Ore

**DOI:** 10.3390/molecules25215028

**Published:** 2020-10-29

**Authors:** Kalu Samuel Ukanwa, Kumar Patchigolla, Ruben Sakrabani, Edward Anthony

**Affiliations:** 1Centre for Thermal Energy and Materials, School of Water, Energy and Environment, Cranfield University, Cranfield MK43 0AL, UK; Kalu.Ukanwa@cranfield.ac.uk (K.S.U.); b.j.anthony@cranfield.ac.uk (E.A.); 2Cranfield Soil and Agrifood Institute, Cranfield University, Cranfield MK43 0AL, UK; r.sakrabani@cranfield.ac.uk

**Keywords:** activated carbon, activating agent, agricultural residues, microwave activation, oil palm waste, trona ore

## Abstract

This study explores the use of a novel activating agent and demonstrates the production and characterisation of activated carbon (AC) from a combine palm waste (CPW) in 3:2:1 proportion by weight of empty fruit bunch, mesocarp fibre and palm kernel shell. The resulting biomass was processed by a microwave-assisted method using trona and compared with material produced by conventional routes. These results demonstrate the potential of trona ore as an activating agent and the effectiveness of using a combined palm waste for a single stream activation process. It also assesses the effectiveness of trona ore in the elimination of alcohol, acids and aldehydes; with a focus on increasing the hydrophilicity of the resultant AC. The optimum results for the conventional production technique at 800 °C yielded a material with S_BET_ 920 m^2^/g, V_total_ 0.840 cm^3^/g, a mean pore diameter of 2.2 nm and an AC yield 40%. The optimum outcome of the microwave assisted technique for CPW was achieved at 600 W, S_BET_ is 980 m^2^/g; V_total_ 0.865 cm^3^/g; a mean pore diameter 2.2 nm and an AC yield of 42%. Fourier transform infrared spectrometry analyses showed that palm waste can be combined to produce AC and that trona ore has the capacity to significantly enhance biomass activation.

## 1. Introduction

Environmental safety has become an increasing global concern in the last three decades. In developing countries, water pollution due to the presence of heavy metals are a threat to both flora and fauna and especially aquatic life [1], and plants [2]. Water contamination by heavy metals has become an increasing problem due to industrialization [3] and this together with the use of pesticides, fungicides, and manufacture of paints, paper and welding activities can contaminate the environment [4]. These heavy metals can be removed by treatment with activated carbon (AC), which is a carbonaceous adsorbent with enhanced porosity and a very high surface area [5,6]. It can be produced using coal, agricultural residues and other waste biomass. However, waste biomass is an important alternative to coal as a rich carbon source and in particular, the wastes associated with the production of the 76 million metric tonnes (MT) of palm oil are of particular importance as they represent about 20% of the total waste generated in the process line producing palm oil [7]. With the addition of palm trunks, fronds and other residues, the annual global waste generation from oil palm is approximately 500 MT.

The use of residual biomass from the palm oil agro-industry has gained more attention in recent years since it can be converted, through technologies like cogeneration, composting, pelletizing, briquetting, pressing, pyrolysis and enzymatic digestion into value-added products [8]. Palm biomass as a feedstock offers an economic contribution to power generation as it reduces operational cost [9]; given the low-cost of palm waste. Pellets and briquettes from oil palm waste are also potential alternatives to traditional fossil fuels in boilers, furnaces and kilns and has a more wide scale applicability in domestic cooking, central heating systems, water purification [10]. Moreover, the ash derived from palm residue is also a useful supplementary cementitious material [11], serving as partial replacement for Portland cement and clinker waste for concrete [12]. Nonetheless more high value uses are desirable and one of them is AC, because of its wide applicability in water treatment, pharmaceutical, desalination, gas storage due to its porosity [13,14].

Low-cost options for AC production, include cassava peel, rice husk, corn pods, bamboo, oil palm waste (OPW) [15,16]. Others include cotton residues [17], tobacco stems [18], bamboo [19], varieties of flower stalks [20], coconut shell and coir, all of which are also good precursors for AC production [21]. However, OPW is among the lowest-cost feedstock with high carbon content of up to 50% and has an excellent potential for AC production [22,23]. They are also suitable feedstock for gasification, as well as AC production [24] and the production of carbon fibre filaments [25].

Nonetheless, OPW is currently an underutilised wastes in some developing countries, due to the high quantities generated in the process [26], its bulkiness, difficulty in transporting them from plantation to processing plants and their assumed low value. Overall, the utilisation of OPW is poor, despite its use for heat generation in boiler unit of palm processing plant [27]. However, as noted above it is good precursor for AC production [28] and the performance of OPW based AC has been demonstrated in several studies on the adsorption of heavy metals [29,30], dyes [31], organic and inorganic pollutants [28]. All of the OPW components typically contain about 5–6% hydrogen in the bound in the form of cellulose, hemicelluloses and lignin [25]. Lignin in particular, as an organic cross-linked polyphenolic polymer, is highly amorphous and contributes to generating the internal structure of AC [32].

Oil palm residues and oil palm processing waste are abundant in Asia and Africa as an agricultural waste and would have many applications as low-cost adsorbent when processed into AC [33]. They can be processed by conventional direct heating or by microwave-assisted pyrolysis method. Microwave systems are a recent development as compared to conventional routes and can create excellent pore structural development within a short process time [34]. However, the non-uniform sizes and shapes of the material in palm oil biomass represent a major disadvantages for its use over other biomass precursors and contributed to the high pre-production cost of AC [35]. Thus, studies are still being directed on the optimum and more eco-friendly modes to achieve efficient production [36]. OPW research has further gained interest in South-East Asia and Africa, where it is abundantly available and could be utilised as adsorbent AC [33]. Finally, while there are numerous ways of producing AC, tut the two dominate methods are physical and chemical activation [37].

Unfortunately, chemical activating agents frequently associated with excessive costs and in some cases can be associated with secondary contamination. Some studies have considered the use of natural and organic chemicals to overcome the challenge [38]. However, for chemical routes relatively inexpensive materials like ZnCl_2_ are generally considered to be particularly effective but still present a major difficulty due to the need for their effectively removal by washing [35]. Bergna et al. [39] identified the differences between one and two-step activation, and thus, in this study we have used two types of two-step method; a conventional route and a conventional-microwave route.

Trona ore (Na_2_CO_3_·NaHCO_3_·2H_2_O) is a crude soda ash commonly deposited on soil surfaces, and it can be found in some parts of Nigeria, in combination with other compounds such as Na_2_SO_4_, CaCO_3_, CaSO_4_, together with high traces of K_2_CO_3_ and about 14% insoluble substances [40]. Some success with individual palm wastes has already been recorded in activation through microwave method [41]. This method improves the physiochemical properties of AC and yield [42]. It also reduces the preparation period and activates with a uniform temperature [43].

Most AC production use inorganic activating agents; however, considering the environment impact of some of the activating agents [44], plus their cost and limited availability an alternative is necessary. Most AC produced are from a mono-precursor, however, this study investigated the preparation of AC from combined waste of palm kernel shell (PKS), mesocarp fibre (MF) and empty fruit bunch (EFB) using microwave system and demonstrates the viability of crude trona ore, and evaluates the effects of the irradiation power and impregnation ratio (IR).

In recent years microwave has been suggested by several researchers as a technique to reduce time and energy when producing AC from biomass [45,46]. This technique has been demonstrated in several studies but has not reached commercial scale due to various challenges. These include design of suitable microwave reactors, and the poor dielectric properties of biomass and the poor understanding of the interaction mechanisms when using biomass which has hindered large-scale development [35]. Therefore, the use in AC production require appropriate design and material selection and optimisation of the parameters.

Hitherto, the use of trona ore and combined palm waste for AC production has not been reported. Na_2_CO_3_ which is the main component of trona ore is a poor activating agent. However, because trona ore is also a mixture of many salts the multi-interactive influence of its various components can enhance the elimination of volatiles and improve tar decomposition. This study looks at AC produced using microwave and conventional method from combined palm waste at different temperature using trona ore. It also evaluates the effectiveness of trona ore in the elimination of alcohols, acids and aldehydes, with a focus of increasing the hydrophilicity of the resultant AC. The optimum parameters and the capacity trona ore as an activating agent are examined using Fourier transform infrared spectrometry (FTIR), thermogravimetric analysis (TGA) and other techniques including scanning electron microscope (SEM). Here, SEM is used to determine the ability of trona ore to creating a comb-like morphology, while Brunauer, Emmett and Teller surface area (S_BET_) for nitrogen adsorption and desorption determine the total volume and surface area of the resulting AC.

## 2. Results and Discussion

### 2.1. Thermogravimetric Analyses of OPW

TGA results are shown in Figure 1, and indicate that the evaporation of moisture occurs between 90–150 °C, and that below 150 °C there is only 10% weight loss with no significant devolatilization occurring until one reaches temperatures ranging from 200 °C to 450 °C. At higher temperatures the decomposition of the lignocellulosic components in the OPW begins, resulting in the formation of condensable hydrocarbons [47] and from 450–900 °C char is formed, along with the partial formation of tar along with the evolution of gases including CO, CO_2_ or H_2_.

Normally, every form of activation process starts at a minimum of 400 °C and the duration depends on the heating medium, quantity and type of material. Here, the TGA for samples impregnated by trona underwent thermal breakdown at 500 °C.

Trona appears to initiate depolymerisation of the biomass as lignin interface with the CO_2_ produced from the thermal breakdown of the activating agent. The cleavage of the numerous intermolecular bonds with hydrogen in the cellulose is due to the release of water [48]. Further treatment at temperature beyond 400 °C yields phenolic monomers and materials like 4-ethylguaiacol. The crystalline structure of cellulose at high temperature retains its morphology in the presence of Na_2_CO_3_. Moreover, hydrophobic interaction is minimised due to the existence of Na_2_CO_3_ precipitates. When hemicellulose undergoes hydrolysis, the changes observed depend on temperature variation and pH [49].

Lignin linkages can also be catalysed by salts due to the presence of ether group. This can result in the release of CO^2−^, Cl^−^ and other ions, which react with unstable monomers, liberating soluble phenolic compounds. However, the resulting bond breakage results in the increase in the hydrophilicity. At temperature of above 400 °C, there is also an increase in the degradation of oligomers and monomers. The rate of gasification is influenced as salt acts as a catalyst to cause cellulose degradation at temperature of about 400 °C. The degradation further produces hydroxycarboxylic acids and the formation of glycolic and lactic acids, along with the increasing generation of low molecular mass fragments at higher temperature.

The addition of trona increases the carbon yield from 6% to 50% and reduces the temperature for the maximum degradation rates as shown in Figure 1. This is because organic metal salts have an effective catalytic effect on char formation [50]. In the presence of only Na_2_CO_3_, there would be no devolatilisation at lower temperature but due to the presence of other salts as impurities, devolatilisation can occur at 300 °C. In the DTA curve, the peak of the weight loss is significant at 450 °C. The moisture content also increases because of salt addition. When trona is added to lignin, the weight loss behaviour slows, and the degradation rate is directly related to the concentration of trona. Typically, K_2_CO_3_ suppresses the formation of char but with trona, the rate of char formation is high. The temperature at 350 °C, and above causes decomposition of cellulose which results in a sharp weight loss.

The addition of activating agent, basically, increase in temperature relative to the duration affects pore development, at 400–600 °C, the volatile displacement is low and tar formation which results in clogging of pores was obvious. The increase in process temperature to about 700 °C causes a rapid release of volatiles and widens the pores. High temperature for a long time, however, lead to low yields and pore structural collapse as observed in the two-step activation process. There is always a period in the process, after the peak of activation, where heat energy is being supplied but there is no change in surface chemistry and morphology. However, beyond this period, the yield begins to decrease as increasing amounts of ash are produced. As the heating duration increases the pores in the AC also gradually begin to collapse.

The moisture content influences the energy requirement for thermal decomposition for both the conventional and microwave activation processes. Microwave offers control on the spatial thermal profile of both the reactor and the material relative to the heating rate (Figure 1). Its presence improves the ability of the process to overcome thermal gradients within the reactor between the main feedstock and the evaporating materials.

Figure 2 shows that the moisture content decreases with increase in microwave heating time. The rapid temperature change between 50–60 °C is due to an increased molecular interaction of the material. The first 120 s is particularly significant and could be used to predict this interaction. The results show that the heat input parameter affects the yield relative to moisture content and gas yield during the entire thermal processes this knowledge is helpful in the design of material requirement for a defined quantity of product.

### 2.2. Elemental Analysis of the Combine Palm Waste Activated Carbon

The elemental analyses of the CPW and their resultant AC are summarised in Table 1. The increased in temperature produced AC with oxygen containing functional group removed from carbon skeletal due to the release of volatiles. During the carbonisation and activation processes decarboxylation and aromatisation results in the breakage of unstable chemical bonds in the carbon matrix, causing the release of volatiles which results in a high fixed carbon (FC) content.

During activation of CPW, the FC content increased from 16.25 wt% to 73.56 wt% for the single step conventional technique and to 76.60 wt% for a 2-step conventional technique, demonstrating the superiority of the two-step process. By contrast the microwave approach can produce material with a fixed carbon content of 75.9%. Therefore, a single step microwave technique produces almost the same result as a two-step conventional process. Reduced volatile content in two-step conventional and microwave processes is due to an increased thermal energy which enhances the evolution of volatiles from the carbon matrix. Although conventional activated ACs have lower volatile content, the hydrogen-carbon ratio is higher than the conventional AC by 0.1–0.5 across this study. This may be due to the microwave irradiation which produces uniform structural heating with low thermal gradient between the outer and interior of the biomass structure.

The sulphur content was thermally stable in both processes and there was no residual sulphur due to the use of trona. There is also an enrichment in carbon content and decline of hydrogen and oxygen content after activation due to the removal of moisture. However, the presence of moisture in the precursor ensured the maintenance of uniform temperature profile during the activation process.

### 2.3. AC morphology and Surface Chemistry Mechanism

The surface morphology of raw PKS, MF, and EFB are illustrated in Figure 3. Activation with trona ore after the evaporation of the volatiles clearly show the creation of well-developed pores. The SEM diagram indicates no major difference due to activation temperature but show clear variation with activation methods. The structure of samples activated by conventional methods look rough and uneven while the samples produced by microwave method appear smoother. The changes in Figure 2 show the difference in surface porosity prepared with each of the OPW. These cavities were created during the thermal process as starch and the organic materials are transformed into volatile and leaving carbon structure behind. The impregnation accompanied by thermal treatment resulted in degradation of the microstructure.

The surface areas of all the samples were measured, and the attenuated total reflectance and absorbance and transmittance were analysed and for better representation, transmittance was used for the comparison. The feedstock for each of the CPW components were also examined along with the correspondent ACs.

For raw feedstock, spectra at 3000–3500 cm^−1^, O-H stretching showing the presence of alcohol, phenol or carboxylic acid which is typical of biomass, and these are eliminated by the activation process. At the range of 1700–1750 cm^−1^, C=O stretching indicates the presence of aldehyde, ketone or carboxylic acid. Low temperature activation shows peak at 2800–3000 cm^−1^, C-H stretching alkanes, and this suggests the need for additional heating and further release of volatiles and biomass tars. In the high temperature range of 800–1200 °C, we can see C-O bending, and reduced alcohol and ether contents. C-H bending signifies the existence of alkanes, which are eliminated by the presence of an activating agent and longer heating duration. Further increase in temperature and activation duration show increased C=O stretching representing the presence of alkenes and aromatic rings.

### 2.4. Effect of Process Parameters and Modes of Production on AC Characteristics

The results presented in Table 2 show that, at 800 °C the S_BET_ are high with high pore volume. However, the yield reduces with increase in temperature. Here, the result indicates that pore volume is linearly proportional to S_BET_ and inversely proportional to pore diameter.

The effect of microwave power in AC production process is obvious as it influences the duration of activation and the carbon yield. An increase in microwave power from 360 to 600 W increase the porosity and adsorption efficiency. S_BET_ and other parameters also increases with an increase in microwave power until the optimum range is achieved. In this analysis, the optimum microwave power range is 600 W. Beyond this range, energy is wasted without corresponding increase in porosity. AC activated using microwave had the highest S_BET_ for CPW, however, the pattern of adsorption remains the same.

One-step production is normally considered to be more energy friendly than the two-step process and will have a higher yield. However, the microwave also offers a more energy friendly route and the low energy requirement of microwave technique is based on the activation duration and fast uniform heating. Possible breakdown of the pore structure is also more common in the two-step process. By contrast, relative trapping of the activating agents in the pores are minimal with microwave heating. This is because microwave heats from interior which is the opposite for conventional heating. Microwave approaches thus saves energy and time; however, it should be noted that biomass is a poor microwave absorber and requires the presence of the right activating agent to improve dielectric properties. AC produced by the microwave assisted process, also has a more defined pore structure and this could be due to rapid escape of volatile from the interior of the material.

Considering the surface chemical and stability of product, this analysis shows that activating agent effect on the biomass prior to activation is significant in terms of the elimination of hydroxyl groups, depolymerisation and degradation to simple sugar and subsequent reaction at the temperature above 600 °C. From Figure 4, shows the differences in the production processes. For PKS samples, process C, has the highest S_BET_ and the highest yield. The highest micropore volume was observed in PKS process A, while that in the PKS process B was the lowest. For the MF, process C has the highest S_BET_, yield and micropore volume. For EFB, process A has the highest S_BET_ and micropore volume, while process D offered the highest total pore volume, but process C had the highest yield.

The CPW sample has a different trend from the other individual feedstock. CPW process C has the highest S_BET_ and total pore volume. CPW process A has the highest mesopore volume while CPW process C and CPW process D has the same micropore volume. For comparison, a study by Hussaro [51] showed that at 700 °C activation temperature, PKS impregnated with Na_2_CO_3_ yielded an AC with S_BET_ of 725 m^2^/g and V_total_ of 0.404 cm^3^/g, and the same technique with ZnCl_2_ resulted in, material with 533 m^2^/g and Vtotal of 0.300 cm^3^/g.

The particle size effect on the individual OPW does not appear to be a factor except for PKS. The process time for MF and EFB also does not depend on the particle sizes. This is the case for both conventional and microwave methods. The optimum temperature for CPW is 800 °C for 1 h, but this depends on the combining ratio of the OPW. With the increase in the quantity of PKS the process time increases. For two step activation, the optimum range is 500 °C and 800 °C for 1 h each. While the one-step process at 800 °C also requires 1 h. This result is based on batch process; however, in a continuous process there would likely be some differences. S_BET_ increases with production temperature, however, it’s also influenced by particle size before and after activation. Initial carbonisation temperature also has direct influence on the displacement of volatiles and thereby affects tar deposition [52]. The N_2_ adsorption–desorption isotherms of AC, at 400–800 °C temperature and activation mode suggest that the AC contain mostly micro pores, which is also shown by the absence of hysteresis.

Although the activation duration is affected by type of reactor, it also depends on other primary factors. In particular, the type of feedstock, affects the time needed to initiate carbonisation and ensure adequate release of volatiles which are; PKS > CPW > MF > EFB. For the microwave method, the time difference is insignificant rather relative to moisture content and microwave power determines the duration of activation [35]. For the conventional production technique, temperature has significant impact in the carbonisation and activation duration. Particle sizes interestingly affect the duration of activation, and the finer the feedstock the shorter the activation duration. Particle size effects were clearly present for PKS, but not for MF and EFB.

### 2.5. Overview of the Process and Resultant Challenges in the Use of Microwave

The primary efficiency of AC depends on the type of feedstock and production techniques. Tar deposition, which results in pore blockages, is one of the challenges preventing AC efficiency. Tar formation is common for all types of biomass pyrolysis but is minimised by operating at a higher temperature, which helps to remove primary tar vapours. This ensures the importance of activating agents which improve tar elimination and widen the AC pores. For OPW and related biomass materials, dimethoxy phenol, trimethoxybenzene and hydroxyl methoxyl benzoic acid can be eliminated at temperature above 250 °C. However, tertiary tars such as naphthalene, acenaphthylene, phenalene and pyrene can create a process challenge by being re-adsorbed by the AC [35], thereby lowering the efficiency of the AC produced [53]. In microwave activation, the dielectric properties of the feedstock are critical for transforming electromagnetic energy into heat. One of the major challenges with most biomass materials including OPW, is their low microwave absorbing properties and the fact that they are weak absorbers with average loss tangent less than 0.1 [54]. Activating agent acts as an adjuvant to increase the microwave absorbability. Furthermore, microwave technique is nearing commercial stage which means that this technique has considerable promise. However, the challenges of developing suitable reactors and appropriate safety measures remain areas which deserve significant attention.

### 2.6. The Effectiveness of Trona Ore

Due to the release CO_2_ from trona due to thermal decomposition above 300 °C, contributes to the activation process. As temperature is further increases the CO_2_ reacts with the char to produce CO; decreasing the overall carbon yield as shown below. Therefore, prolonged activation time is not desirable when trona is used as an activating agent. The carbonisation temperature is very significant in AC production, in this study, only two temperatures were considered, namely 600 and 800 °C. Trona can cause degradation of glucose due to the presence of Na_2_CO_3_ and possibly affect other alkaline substances present. Trona also causes solvation of ether linkages because of delignification, which is a factor in pore creation [35] This further results in the dissolution of xylan, breakage of longer fibres, exfoliation and eventual removal of xylan as outlined in Figure 5 [55].
(1)$${\mathrm{Na}}_{2}{\mathrm{CO}}_{3}+C\;\to -\mathrm{COONa}+\;-\mathrm{CONa}$$
(2)−COONa+C → −CONa+CO 
(3)$$-\mathrm{CNa}+{\mathrm{CO}}_{3}\;\to \;-\mathrm{CONa}+\mathrm{CO}\;$$
(4)$$-\mathrm{CONa}+{\mathrm{CO}}_{3}\;\to \;-\mathrm{COONa}+\mathrm{CO}\;$$

The use of Na_2_CO_3_ as an activating agent also has a significant effect on CO formation at temperature above 600 °C due to alkali lignin gasification [56], although this behaviour is affected by the presence of oxygen. The catalytic effect of trona due to the presence of its two major components: Na_2_CO_3_ and K_2_CO_3,_ is also effective in cellulose pyrolysis [56].

For conventional method, the effect of impregnation ratio is obvious. At 1:0.5 impregnation the pores are not well defined; however, they became more developed at 1:1. At 1:2, the structural walls appear to collapse. For microwave activation method, the walls were intact and well defined for 1:0.5. S_BET_, which is an important factor for adsorption depends on the particle size of the AC. A comparison of trona ore AC with other activating agents is shown in Figure 5. The pore sizes of AC activated by trona ore indicate good characteristics. Most AC are used for water purification; and should contain no trace of corrosive or toxic chemicals. Hence, activation with trona makes it easy to wash the AC, but still guarantees its safety if improper washing fails to remove the trona.

To ensure the efficiency of CPW, the individual OPW must be of similar particle sizes. For individual OPW, the process is less complex and gives a more easily predictable outcome. However, an individual OPW may have challenges such as poor carbon yield for MF and EFB, therefore complementing them with PKS can increase the yield in a single process. However, thorough mixing and stirring during impregnation are important for uniforms result. For CPW, the individual components must also be properly mixed before impregnation and impregnation duration could increase.

## 3. Materials and Methods

### 3.1. Materials

The feedstock was selected from a processing mill at the palm plantation site of Desai Impex Nigeria Ltd. in Abia state, Nigeria. It was first ground into a particle sizes of about 0.5–2 mm using a hammer mill then it washed with warm water 60–80 °C to remove oil residues and finally rinsed with de-ionised water. The OPW samples were dried for five days and supplementary dried in an electric oven at 70 °C for 4 h, to eliminate surface moisture. Trona ore was obtained from Desai Impex Nigeria Ltd. where it is used for cleaning oily vessels. De-ionised water was used throughout the experiment and chemicals were of analytical grade. For CPW, the individual wastes were 3:2:1 proportion by weight of EFB, MF and PKS proportion by weight respectively. This combination gave a more balanced and uniform blend for this study.

The analysis for individual palm wastes [57,58] are given in Table 3. This shows the differences between the individual feedstock. The high carbon content of the three individual waste qualifies them as appropriate feedstock for AC production.

The thermal processes were done under a controlled condition; the overall carbon conversion was evaluated. Carbon conversion efficiency $\eta \;\left(\%\right),$ the volume of volatile production is calculated based on Equations (5) and (6) respectively.
(5)$$\eta \;\left(\%\right)=(1-\frac{\mathrm{Carbon}\;\mathrm{in}\;\mathrm{residue}\;\left[\frac{\mathrm{mol}}{S}\right]}{\mathrm{Carbon}\;\mathrm{in}\;\mathrm{feedstock}/\mathrm{fuel}\;\left[\frac{\mathrm{mol}}{S}\right]})\times \;100$$
(6)$$V_{\mathrm{gas}}\left({\mathrm{cm}}^{3}\right)=\frac{W_{\left(0-f\right)}{\mathrm{Xx}}_{c}}{\sum x_{t}\;(\frac{{\mathrm{MW}}_{c}}{V_{\mathrm{STP}}})\times \eta }$$
here, the volume of gas collected at the end of the process (m^3^). The final weight is denoted by W_f_ and the initial weight of CPW sample is W_0_. The post process weight is represented by W_(0-f)_ (kg), η is the conversion efficiency of carbon which is the ratio of AC produced and biomass available. Xi is the product gas volume. V_STP_ is the volume of 1 mol of ideal gas at STP. x_c_ is the carbon content. MW is microwave radiation measured in power.

Brief descriptions of the scenarios for analysis procedures were performed by simulation. Evaluation for mass, energy balances were outlined. Aspen plus V.10 under the academic license was used for pyrolysis. Activation process in relation to material and energy balances were simulated using the background Equations (7)–(16). Pyrolysis and activation conditions were formulated relative to the gas’s emissions and analysed and the macro thermogravimetric profile during the thermal decomposition was recorded.

Mass balance of activation process is based on the summary available elsewhere [59].
(7)$${\mathrm{Mass}}_{\left(g\right)}=\mathrm{Char}+\mathrm{gases}+\mathrm{Water}+\mathrm{Tars}+\mathrm{condensable}$$

For gases:(8)$$m_{\mathrm{gases}}=\sum m_{\mathrm{gas},\;i}$$
(9)$$V_{\mathrm{gas},i,\;t}=\sum_{0}^{\mathrm{tf}}V_{\mathrm{gas},i,\;t}$$
(10)$$\%\;\mathrm{Material}\;\mathrm{yield}={Y\%}_{\mathrm{AC}\;\mathrm{product}}=\frac{m_{\mathrm{AC}\;\mathrm{product}}}{m_{\mathrm{biomass}}}\times 100$$

Energy input based on solid material is calculated based on the Lower Heating Value on a dry basis (LHV_dry_) (kJ/kg or kJ/Nm^3^).

For energy input:(11)$$E_{i,\mathrm{biomass}}={\mathrm{LHV}}_{\mathrm{biomass}}\left(\frac{\mathrm{kJ}}{\mathrm{kg}}\right)\times m_{\mathrm{biomass}}$$
(12)$$\begin{array}{ll} E_{i,\;\mathrm{process}}= & m_{\mathrm{biomass}}\left(\mathrm{kg}\right)\times C_{p\;\mathrm{wood}\left(\mathrm{lower}\;\mathrm{activation}\;\mathrm{temp}-20^{{}^{\circ}}C\right)}\times \Delta \theta \\ & \;+m_{\mathrm{AC}}\left(\mathrm{kg}\right)\times C_{p,\;\mathrm{AC}\;\left(\mathrm{higher}\;\mathrm{activation}\;\mathrm{temp}-\mathrm{lower}\;\mathrm{activation}\;\mathrm{temperature}\right)}\times \Delta \theta \end{array}$$

The output energy (E _output_ (kJ) is based on the quantification of AC, dried gas, condensable and activation gases. These are outline below:(13)$$E_{0,\;\mathrm{AC}}={\mathrm{LHV}}_{\mathrm{AC}}\left(\frac{\mathrm{kJ}}{\mathrm{Nitrogen}\;\mathrm{litre}}\right)\times m_{\mathrm{AC}}\;\left(\mathrm{kg}\right)$$
(14)$$E_{0,\mathrm{dried}\;\mathrm{gas}}=\sum E_{\mathrm{gas},i}$$
(15)$$E_{\mathrm{gas},\;i}={\mathrm{LHV}}_{\mathrm{gas},i}\left(\frac{\mathrm{kJ}}{\mathrm{Nitrogen}\;\mathrm{litre}}\right)\times V_{\mathrm{cummulated}\;\mathrm{gas},\;i}\;\left(\mathrm{Nitrogen}\;\mathrm{gas}\right)$$
(16)$$E_{0,\mathrm{activation}\;\mathrm{gases}}=E_{0,\mathrm{dried}\;\mathrm{gases}}+E_{0,\mathrm{condensable}}$$

The microwave construction and set-up were done by Industrial microwave systems, Milton Keynes, UK. The device is a modified system from a domestic microwave Panasonic NN-SD27HS (Panasonic U.K. Ltd., Berkshire, UK). It was fitted with vents and reconstructed to allow an efficient pyrolysis process. Teflon tubes were used as delivery lines in all the processes as illustrated in Figure 6. For efficient delivery and management of the volatiles, Teflon pipes were constructed to support inert gas supply and volatile outlet. The outlet is affix to a condenser and a pressure regulator (Cranfield University, UK). The safety of the system depends on the management of material handling, pressure regulation and establishing a zero-microwave leakage. The production was done at 460, 600, 800 and 1000 W at different microwave times ranging from 10–20 min.

### 3.2. Production of Activated Carbon

The OPW ground was introduced into a beaker mixed with trona ore dissolved in water, then stirred to form a solution, 1 g/mL. The samples were subject to an impregnation ratio of 0.5:1, 1:1, 1:2 and stored for 18 h in a fume cupboard (Cranfield University, UK). There is also a control sample without an activating agent. The sample was carbonised in a high temperature laboratory electric furnace (Carbolite, Hope valley, UK) at 500, 600, 800 °C for 1 and 2 h. Another set of samples was put in the microwave and power was varied between 450 and 1000 W. A 200 cm^3^/min flow rate stream of pure nitrogen gas (99.9%) was used throughout the production process. After cooling in a desiccator, the samples were soaked and washed in de-ionised water until filtrate reached a neutral pH. Materials were then filtered and dehydrated in the oven at 105 °C for 18 h to constant weight. The carbonised sample was ground with a 15 horsepower biomass hammer mills (HM10) (Ecostan, Punjab, India) to an l size of 0–2 mm with standard sieve and kept in a desiccator for about 2 h to ensure no adsorption of moisture. The two-step process involves the production of biochar and further treatment of the biochar with activating agent to produce more efficient AC.

### 3.3. Characterisation of Trona Ore Activated Carbon (ToAC)

The thermal properties and behaviour of OPW were evaluated before and after impregnation with trona ore using thermogravimetric analyser (Perkin Elmer Pyris 1) (PerkinElmer, Shelton, WA, USA). The analyser was programmed for 4 h and the analysis of the behaviour was evaluated based on weight loss relative to time. ToAC structure was observed individually using SEM (Phillips XL30ESEM) (SEMTech Solutions, Billerica, MA, USA) magnified between 500–2500 times that of the original size. Surface texture relative to activation temperature and mode were studied. The resultant images were compared, the pore symmetry and sizes were determined. The S_BET_ and material structural porosity of the ToAC were evaluated for micropore, mesopore and macropore by N_2_ adsorption/desorption isotherm using a surface area analyser (3P surface DX, 3P Instrument GmbH &Co, Odelzhausen, Germany). This is based on the principle of relative pressure and adsorbed volume. Fourier transform infrared (FTIR) (Bruker, VERTEX 70,, Ettlingen, Germany) analysis of the AC was done to assess and identify organic, polymeric and inorganic compounds in the material. These techniques were also used for the surface chemical comparison, relationship using infrared emission and adsorption spectra. It determines the infrared adsorption bands, identify the molecular components and structures. These were analysed using OPUS software linked device (Bruker, VERTEX 70, Ettlingen, Germany) (FTIR) in Cranfield University UK. The thermogravimetric analysis is used for macroscopic kinetic modelling by evaluating the temperature change relative to mass loss. Therefore, in this study all pyrolysis products are categorised as char, tar and gases.

## 4. Conclusions

Palm waste in a combined form can be an effective feedstock for the production of AC with trona ore. The morphology of the AC produced were honeycomb-like, the total volume and surface area of the AC produced by this technique improved relative to the conventional approach. The AC yield was improved in this process and this confirms that direct activation process allows an effective activating process for the release of volatiles with no obstruction of the pores.

The method of activation influenced the morphology, a single-step activation with microwave has more unbroken pores while the two-step and conventional techniques produce ACs with broken and uneven pore linkages. The microwave activation method was better due to energy and time saving due to a reduced activation duration. The pore structures were enhanced by the increase in microwave power and temperature. The AC produced in this study has the potential to be used for heavy metal, dyes and other gases adsorption given their high surface area of about 1220 m^2^/g, and well-defined pore structures.

## Figures and Tables

**Figure 1 molecules-25-05028-f001:**
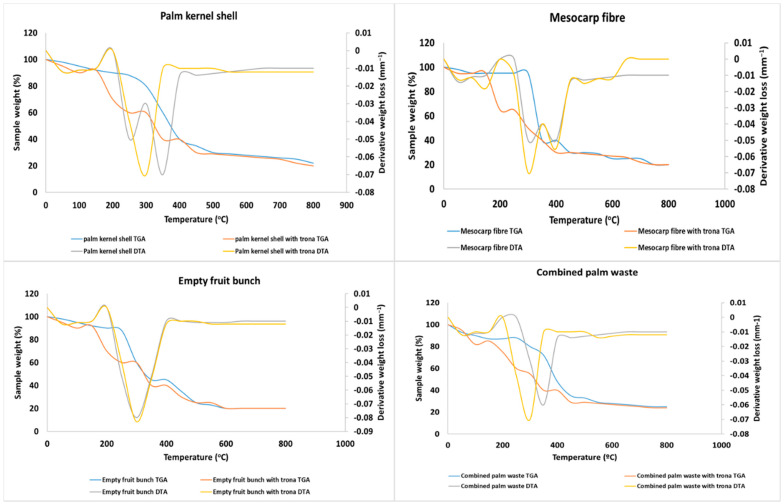
DTA/TGA of OPW and variation of the samples impregnated with trona ore.

**Figure 2 molecules-25-05028-f002:**
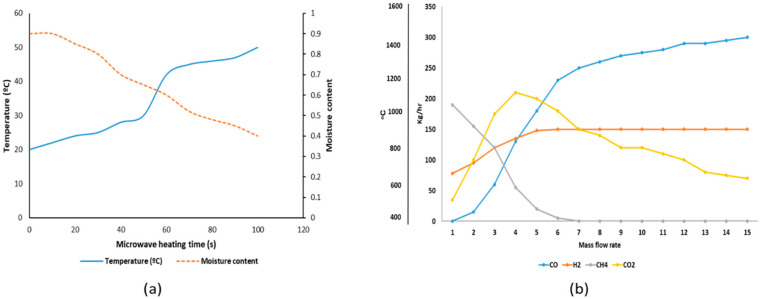
(**a**) Microwave heating time relative to moisture content and temperature variation. (**b**) Mass flow relationship relative to the influence of temperature change on the composition of syngas.

**Figure 3 molecules-25-05028-f003:**
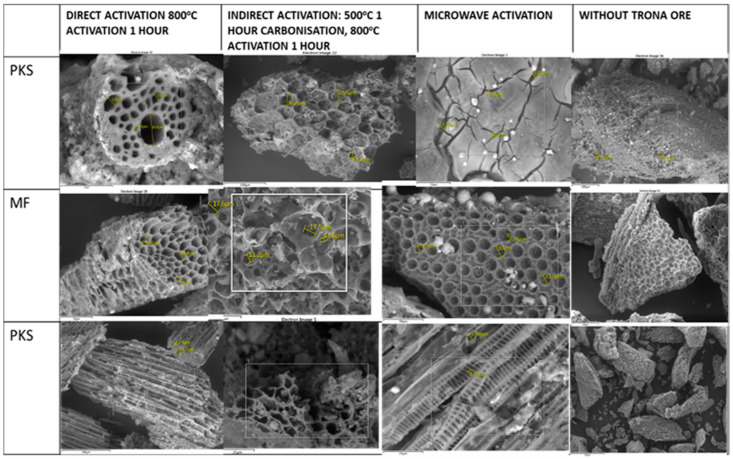
SEM analysis of the AC produced from trona.

**Figure 4 molecules-25-05028-f004:**
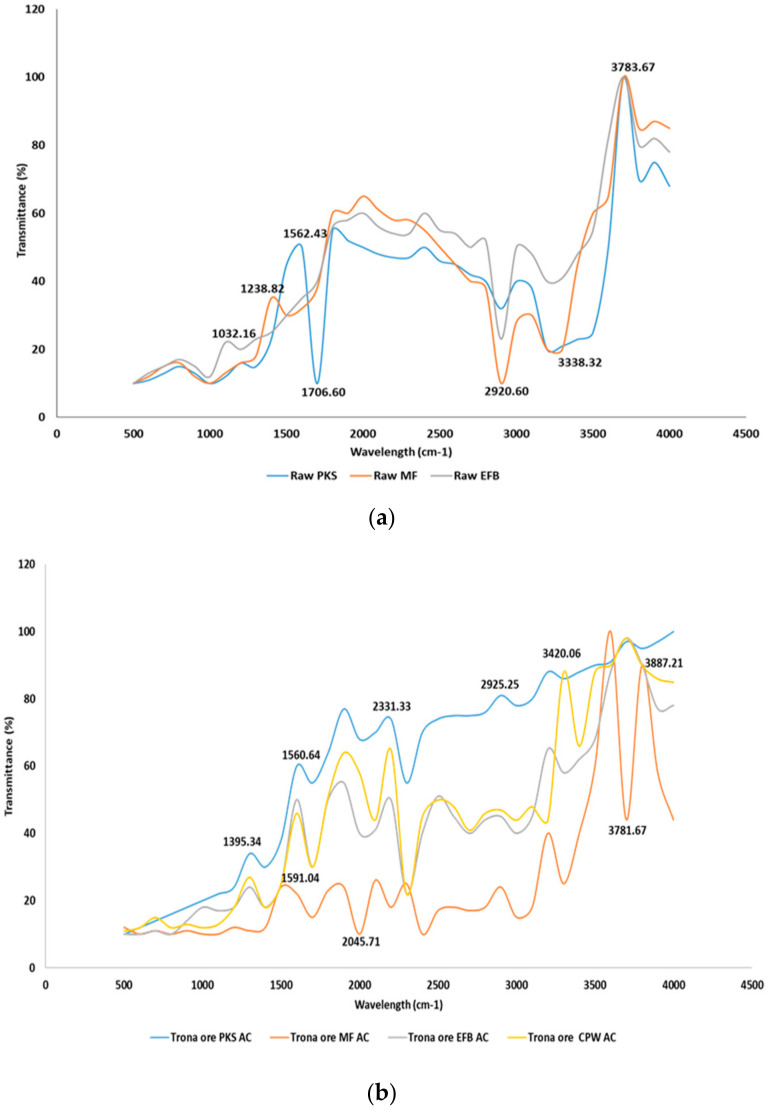
FTIR of OPW from different processing methods and activation conditions (**a**) raw feedstock (**b**) trona ore activated AC.

**Figure 5 molecules-25-05028-f005:**
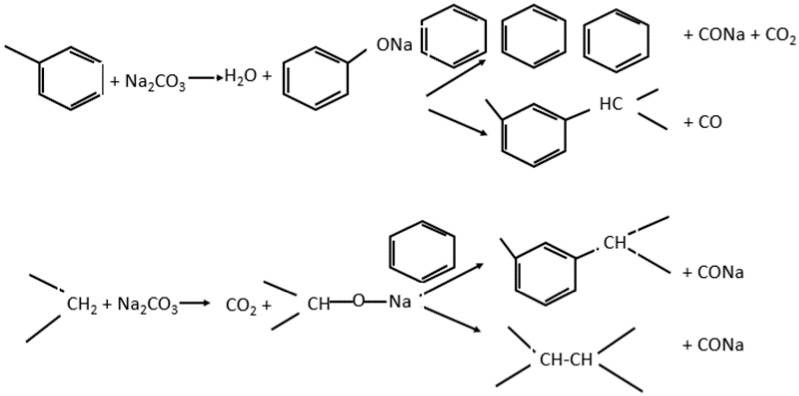
Mechanism of trona during activation process.

**Figure 6 molecules-25-05028-f006:**
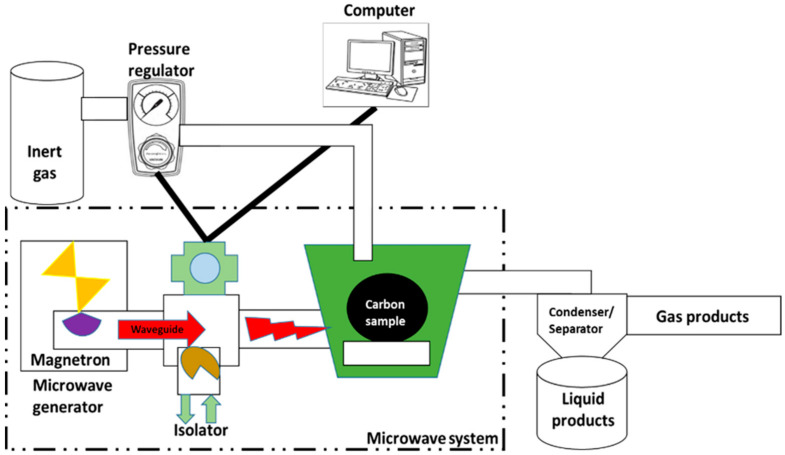
Microwave experimental setup for AC production.

**Table 1 molecules-25-05028-t001:** Elemental analysis of combine palm waste activated carbon based on different production processes.

Sample/Process Parameters	Proximate Analysis				Ultimate Analysis				
	Moisture	Volatile	FC	Ash	C	H	N	O	S
Raw CPW	4.05	74.50	16.25	5.20	44.60	6.35	0.80	48.10	0.15
CPW ℗ A: 800 °C-1 h	5.25	12.95	73.56	8.24	74.20	2.88	1.10	21.70	0.12
CPW ℗ B: 500 °C-1 h + 800 °C-1 h	2.46	10.38	76.60	10.56	76.95	2.22	1.38	19.33	0.12
CPW ℗ C: 600 W-20 min	6.44	14.69	71.22	7.65	73.87	3.10	1.30	21.63	0.10
CPW ℗ D: 500 °C-1 h + 600 W-10 min	4.05	11.20	75.90	8.85	74.63	2.42	1.18	21.67	0.10

CPW: combine palm waste, ℗: Process.

**Table 2 molecules-25-05028-t002:** Yield and pore structural parameters.

Sample	Process Parameter	S_BET_ (m^2^/g)	V_total_ (cm^3^/g)	V_meso_ (cm^3^/g)	V_micro_ (cm^3^/g)	D_p_ (nm)	Yield (%)
PKS	℗ A: 800 °C-1 h	923	0.750	0.122	0.285	3.2	35
MF		1105	0.882	0.230	0.302	3.4	42
EFB		845	0.645	0.234	0.285	2.3	30
CPW		920	0.840	0.356	0.354	2.2	40
PKS	℗ B: 500 °C-1 h + 800 °C-1 h	650	0.745	0.108	0.230	2.8	30
MF		736	0.646	0.280	0.262	2.4	30
EFB		820	0.568	0.250	0.145	2.5	24
CPW		870	0.622	0.286	0.162	2.0	34
PKS	℗ C: 600W-20 min	1030	0.825	0.105	0.245	3.3	42
MF		1220	0.887	0.274	0.465	3.8	45
EFB		735	0.640	0.222	0.346	3.1	37
CPW		980	0.865	0.256	0.380	3.3	42
PKS	℗ D: 500 °C-1 h + 600W-10 min	670	0.542	0.089	0.200	3.1	37
MF		864	0.650	0.182	0.230	2.8	28
EFB		810	0.712	0.234	0.242	3.0	28
CPW		900	0.660	0.310	0.380	3.0	38

PKS: palm kernel shell, MF: Mesocarp fibre, EFB: Empty fruit bunch, CPW: combine palm waste, V: Volume, Dp: Pore diameter, ℗: Process.

**Table 3 molecules-25-05028-t003:** Biochemical analysis of individual palm waste biomass.

		PKS	MF	EFB
Proximate analysis (%*w*/*w*)	Moisture	12	12.1	14.4
	Ash	1.5	4.8	4.4
	Volatiles	70.6	72.9	73.7
	Fixed carbon	15.9	10.5	7.5
Ultimate analysis (%*w*/*w*)	C	46	45.8	37.5
	H	5.1	6.3	5.0
	N	0.4	0.9	0.4
	S	0.02	0.2	0.1
	O *	35	29.5	38
Lignocellulosic composition **	Cellulose	20.8	33.9	38.3
	Hemicellulose	22.7	26.1	35.3
	Lignin	50.7	27.7	22.1
Thermal and energy properties	Organic content	94.2	92	95.7
	Inorganic content	5.8	8	4.3
	Combustion rate, C_R_ (X × 10^−8^ kg/s) **	4	4.2	3.8
	Specific Heat, c, (J/kgK) **	3113	3231	2832

* Oxygen by difference include moisture and ash, ** [58], PKS: palm kernel shell, MF: Mesocarp fibre, EFB: Empty fruit bunch.

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
