# Peer review of "Preparation and Characterisation of Activated Carbon from Palm Mixed Waste Treated with Trona Ore"

_molecules, 2020, doi:10.3390/molecules25215028_

Round 1
Reviewer 1 Report
The paper Preparation and Characterization of Activated Carbon from Mixed Waste treated with Trona ore by Kalu Ukanwa at all. need some improvements. First of all there are several instances where the language is not clear . For example in line 30 after pollution due to the presence of these heavy metals and there was no previous reference to any metals. In line 32 The use of pesticides, fungicides, and manufacture of paints, paper and welding activities can contaminate the environment due to heavy metals suggests that the pesticides can contaminate environment due to heavy metals in the environment, similar line 64/65 :All of the OPW components typically contain about 5-6% hydrogen 5-6% in the form of cellulose, hemicelluloses and lignin a suggests that hydrogen is in the form of cellulose not a part of the structure of the cellulose. The text should be checked and such clarification introduced.
Second I would like to have clear comparison of effects of pure Na2CO3, pure Trona ore and Trona ore from Nigeria with many other salts on the process of activation to have conclusions in line 149 supported.
Similarly the elementary analysis ( C,O,H) content of the obtained activated carbon shall be presented for comparison with other properties of the activated carbon.
The effects of time of carbonisation and activation and temperatures of the process on the parameters of the product shall be studied in more details and be presented for the readers.
Author Response
The paper Preparation and Characterization of Activated Carbon from Mixed Waste treated with Trona ore by Kalu Ukanwa at all. need some improvements. First of all there are several instances where the language is not clear.
Thank you for your careful observations and remarks. We have made the necessary changes and believe the paper is significantly enhanced.
For example, in line 30 after pollution due to the presence of these heavy metals and there was no previous reference to any metals. In line 32 The use of pesticides, fungicides, and manufacture of paints, paper and welding activities can contaminate the environment due to heavy metals suggests that the pesticides can contaminate environment due to heavy metals in the environment, similar line 64/65 :
Resolved, Reference attached to support claim; Hou et al., 2019 Line 35-36
All of the OPW components typically contain about 5-6% hydrogen 5-6% in the form of cellulose, hemicelluloses and lignin a suggests that hydrogen is in the form of cellulose not a part of the structure of the cellulose. The text should be checked and such clarification introduced.
Restructured and repeated word deleted, please check line 67
Second I would like to have clear comparison of effects of pure Na2CO3, pure Trona ore and Trona ore from Nigeria with many other salts on the process of activation to have conclusions in line 149 supported.
The comparison is very important. However, there are few studies where Na2CO3 was used for activation of palm waste and there is not enough information to allow the production of a separate table. Therefore, we have introduced a statement of comparison from the work of Hussaro et al 2014 to buttress the evaluation, on line 275-278. The data for pure trona is not available therefore the comparison is limited to Na2CO3 and ZnCl2
Similarly the elementary analysis ( C,O,H) content of the obtained activated carbon shall be presented for comparison with other properties of the activated carbon.
Thank you for the suggestion. A new sub-section is included to describe the result. The inclusion has clarified the general objective of the paper. Please, see table 1 according to your recommendation.
The effects of time of carbonisation and activation and temperatures of the process on the parameters of the product shall be studied in more details and be presented for the readers.
We agree more details on time, impregnation ratio and microwave power are necessary and where possible we have added extra information. However, this study’s emphasis is primarily on the influence of the different processes as outlined in Table 2, hence, the optimisation of different process duration was not highlighted.
Reviewer 2 Report
This is an interesting paper applying microwave irradiation (activation) with and without Trone ore in the preparation of AC from palm mixed waste. Results obtained are compared with conventional method. This raw material is widely studied for AC production. There are also some earlier studies related to the use of soda as activator instead of e.g. KOH. Further, microwaves are also applied in several related studies. Author claim the novelty being in the activating agent used. Is it possible to open this novelty statement more detailed? It is only mentioned in the abstract, not in the body text.
The second key question is the added value of this approach compared with the conventional methods. It seems that both methods give quite similar results. Authors claim some issues related to lower energy in the MW treatment in the paper, but it is not verified in this paper. Further, what is the technological readiness level for MW treatment in the production of AC?
Results section contains only characterization results (missing mass/energy balances). Chapter 3 contains also results. This part of the paper should be reconsidered and partly rewritten. It has been earlier observed the difference between one and two-step activation (see example Bergna et al. C: J. Carbon Research 2018, 4, 41.) Some important data is also missing, eg. carbon content and elemental analysis of samples carbonization and activation.
Some minor issues:
Quality of graphical abstract is poor, especially text in the text boxes is too small
Corrections in the references are also needed, see ref. 38 and 53.
Author Response
This is an interesting paper applying microwave irradiation (activation) with and without Trone ore in the preparation of AC from palm mixed waste.
Results obtained are compared with conventional method. This raw material is widely studied for AC production.
There are also some earlier studies related to the use of soda as activator instead of e.g. KOH. Further, microwaves are also applied in several related studies.
Author claim the novelty being in the activating agent used. Is it possible to open this novelty statement more detailed? It is only mentioned in the abstract, not in the body text.
Thank you for your observation, a brief expression of why trona ore was chosen based on safety and cost. The novelty statement is further expressed and included in the introduction..
The second key question is the added value of this approach compared with the conventional methods. It seems that both methods give quite similar results.
Both techniques are good, the major differences are production time and the uniformity of the pore structure. There is evidence of the production of a well-defined structure in the microwave technique. This could be further assessed by adsorption experiments.
Authors claim some issues related to lower energy in the MW treatment in the paper, but it is not verified in this paper. Further, what is the technological readiness level for MW treatment in the production of AC?
The lower energy is relative to process duration. We can understand that production time varies and this justifies the claim about the energy expenditure, given that the microwave technique has a very short production time. The energy quantification was simulated as part of an Aspen plus for economic assessment.
The microwave technique is nearing the commercial stage. However, the challenges of reactors and safety measures remain but we expect them to be dealt with in the near future. This technique is also currently being studied in various projects for large-scale production with a focus on its ability to handle heterogeneous materials (314-317).
Results section contains only characterization results (missing mass/energy balances). Chapter 3 contains also results.
This part of the paper should be reconsidered and partly rewritten. It has been earlier observed the difference between one and two-step activation (see example Bergna et al. C: J. Carbon Research 2018, 4, 41.)
Some important data is also missing, eg. carbon content and elemental analysis of samples carbonization and activation.
The study of mass and energy balances are relevant to the entire project, which was considered in pyrolysis process for the development of the equipment and technology. Based on your recommendation, the activation aspect is included in the methodology and discussion and we have endeavoured to provide more complete information.
Thank you for recommending the interesting work of Bergna et al. which identified the difference between one and two-step activation. Consequently, in this study we have ensured that there are two kinds of two-step methods used. Conventional-conventional and conventional-microwave; thank you for the suggestion. See Line 86-88.
We also considered the AC elemental analysis for just CPW, as being necessary and enough to enlighten more on the aim of the study. A new sub-section is included to describe the result. The inclusion improved the clarity and value of the paper. Please, see Table 1 according to your recommendation.
Some minor issues:
Quality of graphical abstract is poor, especially text in the text boxes is too small
Corrections in the references are also needed, see ref. 38 and 53.
Thank you for another kind observation, the quality of the graphical abstract has been improved.
The Reference is formatted based on Molecules reference style; all errors we believe have been identified and corrected.
Round 2
Reviewer 2 Report
Authors have carefully corrected the paper. I will suggest the acceptance of this paper.